# DUALAUG: EXPLOITING ADDITIONAL HEAVY AUGMENTATION WITH OOD DATA REJECTION

## ABSTRACT

Data augmentation is a dominant method for reducing model overfitting and improving generalization. Most existing data augmentation methods tend to find a compromise in augmenting the data, *i.e.*, increasing the amplitude of augmentation carefully to avoid degrading some data too much and doing harm to the model performance. We delve into the relationship between data augmentation and model performance, revealing that the performance drop with heavy augmentation comes from the presence of out-of-distribution (OOD) data. Nonetheless, as the same data transformation has different effects for different training samples, even for heavy augmentation, there remains part of in-distribution data which is beneficial to model training. Based on the observation, we propose a novel data augmentation method, named **DualAug**, to keep the augmentation in distribution as much as possible at a reasonable time and computational cost. We design a data mixing strategy to fuse augmented data from both the basic- and the heavy-augmentation branches. Extensive experiments on supervised image classification benchmarks show that DualAug improve various automated data augmentation method. Moreover, the experiments on semi-supervised learning and contrastive self-supervised learning demonstrate that our DualAug can also improve related method.

## 1 INTRODUCTION

Deep neural networks lead to advances in various computer vision tasks, such as image classification (Krizhevsky et al., 2012; He et al., 2016; Dosovitskiy et al., 2021), object detection (Girshick et al., 2014; Ren et al., 2015; He et al., 2017), and semantic segmentation (Chen et al., 2017). The training of deep neural networks often relies on data augmentation to relieve overfitting, including recent automated data augmentations which increases the amount and diversity of data by transforming it following some policies (Cubuk et al., 2019; 2020; Lim et al., 2019; Zheng et al., 2022). Applying it normally requires searching for appropriate transformation operators, range of magnitudes, *etc.* Although a moderate level of data transformation can improve model accuracy, heavy augmentation which significantly increases the data diversity sometimes destroys semantic context in the augmented visual data and leads to unsatisfactory performance, as shown in Figure 1 (heavy augmentation part of the red line) and Figure 2(b).

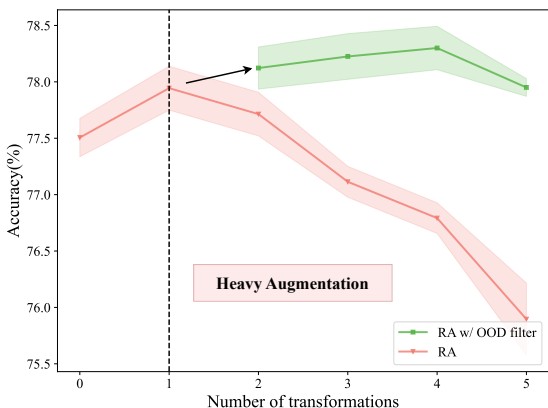

Figure 1: **Red Line**: The performance of RandAugment (RA) peaks with a carefully selected moderate number of transformations, but it rapidly declines if the number is increased further. **Green Line**: After simply filtering out OOD data, the performance of RA further improves even with heavy augmentation. The experiment is conducted on CIFAR-100 using WideResNet-28-2.

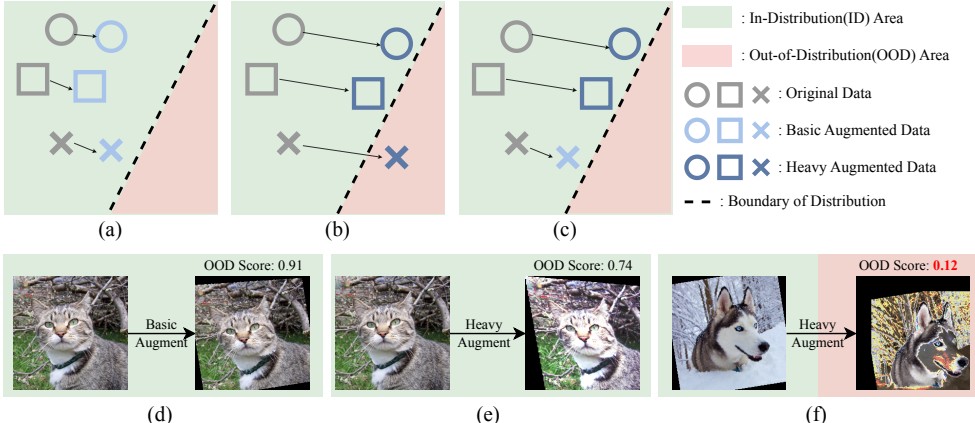

Figure 2: Motivation of DualAug. *Circle*, *Square* and *Cross* represent different data point. **(a)** Current automated augmentations with conservative parameters need to keep the data in distribution thus they are insufficient for data like the *Circle* and *Square* in the figure. **(b)** When heavy augmentation is introduced, the *Cross* become OOD. **(c)** Modify the OOD data (*i.e.*, *Cross*) into in-distribution data appropriately. **(d)** Visualization of basic augmentation, which is not sufficiently. **(e)** Visualization of heavy augmentation, which is more sufficiently (augmented data is in in-distribution Area). **(f)** Visualization of heavy augmentation (augmented data is in OOD Area)

Despite being somewhat effective, a moderate level of transformation in fact augments the whole dataset conservatively, and many training samples may not be sufficiently augmented. See Figure 2 for an illustration of this problem. Comparing Figure 2(a) and Figure 2(b), we can see that, when searching for the optimal data augmentation strategy, there exists a trade-off between the diversity of augmentation and the distraction of semantically meaningless augmentations (shown in Figure 2(f)). From our perspective, the samples without meaningful semantic contexts are OOD samples, hence, in this paper, we consider the possibility of improving diversity without generating OOD samples. A direct idea for achieving this is to detect OOD augmentation results and get rid of them. See Figure 2(c) for expected results.

Based on this intuition, we propose dual augmentation (DualAug), which applies heavy data augmentation while keeping the in-distribution augmentation data as much as possible. DualAug consists of two different data augmentation branches: one is the basic data augmentation branch which is the same as existing augmentation methods (Cubuk et al., 2019; 2020; Zheng et al., 2022), and the other is for heavy augmentation (*i.e.*, more types or larger magnitudes of transformations). We take advantage of the training model for building the OOD detector and we calculate the OOD score according to an estimation of the distribution of augmented data in the basic branch. The threshold for filtering out OOD augmented data is chosen by utilizing the $3\sigma$ rule of thumb. Meanwhile, augmented data that does not meet the $3\sigma$ rule, from the heavy augmentation branch, will be regarded as OOD. These OOD samples are caused by excessive transformation, and they will be replaced with their basic augmentation version to keep the augmentation results in distribution as much as possible. Finally, the in-distribution heavy augmentation results together with some basic augmentation results (which are replacements of the OOD results) will be used in model training.

To summarize, the contribution of this paper is as follows:

- We demonstrate that, although heavy augmentation destroys semantic contexts in some data, there still exists augmentation results which are informative enough and can be beneficial to model training.

- We investigate the existence of OOD augmentation result, and we show that, to achieve better performance, it is important to filter OOD data as much as possible when using heavy augmentation.

- Based on our observation, we propose a two-branch data augmentation framework called DualAug. DualAug makes data further augmented while avoiding the production of OOD data as much as possible.

- Extensive experiments on image classification tasks with four datasets prove that DualAug improve various data augmentation methods. Furthermore, DualAug can also improve the performance of semi-supervised learning and contrastive self-supervised learning.

## 2 RELATED WORK

### 2.1 AUTOMATED DATA AUGMENTATION

Over the past few years, data augmentation and automated data augmentation have developed rapidly (Mumuni & Mumuni, 2022). A pioneer automated augmentation method is AutoAugment (Cubuk et al., 2019), which proposes a well-designed search space and employs reinforcement learning to search the optimal policy. Although it has demonstrated significant performance gains, its high computational cost can pose a significant challenge in certain applications. To address this issue, several different works have been proposed to reduce the computation cost of AutoAugment. Fast AutoAugment (FAA) (Lim et al., 2019) considers data augmentation as density matching, which can be efficiently implemented using Bayesian optimization. RandAugment (RA) (Cubuk et al., 2020) explores a simplified search space that involves two easily understandable hyper-parameters, which can be optimized with grid search. Differentiable Automatic Data Augmentation (DADA) (Li et al., 2020) focuses on utilizing differentiability for efficient data augmentation, thus significantly reducing the cost of policy search. Recently, Deep AutoAugment (DeepAA) (Zheng et al., 2022), a new multi-layer data augmentation search method outperforms all these methods.

Although data augmentation increases the diversity of data, it still faces some problems if not restricted (Wei et al., 2020; Cubuk et al., 2021; Gong et al., 2021; Suzuki, 2022; Wang et al., 2022; Liu et al., 2023; Ahn et al., 2023). For instance, Suzuki (2022) discovers that adversarial augmentation (Zhang et al., 2019) can produce meaningless or difficult-to-recognize images, such as black and noise images, if it is not constrained properly. To address this problem, the proposed method TeachAugment (Suzuki, 2022) uses a teacher model to avoid meaningless augmentations. Although TeachAugment makes reasonable improvements, it has a significant drawback: it involves alternative optimization that relies on an extra model, which significantly increases the training complexity. Our method also handles unexpected augmented data, without introducing any extra model.

### 2.2 OUT-OF-DISTRIBUTION DETECTION

OOD detection is critical in ensuring the safety of machine learning applications (Yang et al., 2021). Hendrycks & Gimpel (2016) uses the maximum softmax probability (MSP) score to detect OOD examples. In ODIN (Liang et al., 2017), the temperature is utilized in the computation of MSP, and it has been demonstrated to be more effective in distinguishing between ID samples and OOD samples. Since then, many studies have been conducted to improve the OOD detection performance in different scenarios and settings (Techapanurak et al., 2020; Liu et al., 2020; Sun & Li, 2022).

It is important to note that OOD detection is a broad concept that has been systematically researched and classified by Yang et al. (2021). Generalized OOD detection includes sensory anomaly detection, one-class novelty detection, multi-class novelty detection, open set recognition, outlier Detection, *etc.* As the first attempt of adopting OOD detection to data augmentation, our DuagAug uses the most basic method for OOD detection (*i.e.*, MSP).

## 3 METHOD

### 3.1 PRELIMINARIES

An image classification model $f$ parameterized by $\theta$ is trained using a training set $D = (x_i, y_i)_{i=1}^{N}$ to correctly classify images. Data augmentation has emerged as a popular technique for addressing the problem of overfitting. The whole data augmentation pipeline can be regarded as a cascade of $M$ individual transformations:

$$\Phi(x, K) = \phi_M \circ \phi_{M-1} \circ \cdots \circ \phi_2 \circ \phi_1 \tag{1}$$

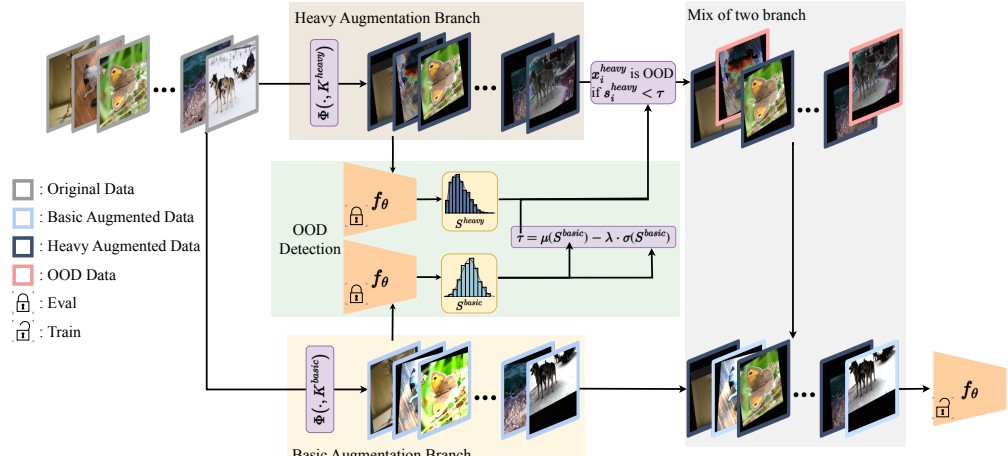

Figure 3: The overview of the proposed DualAug, which has two branches for data augmentation. (a) **Basic Augmentation Branch**: an automated augmentation are chose such as AutoAugment (Cubuk et al., 2019), RandAugment (Cubuk et al., 2020) and DeepAA (Zheng et al., 2022). (b) **Heavy Augmentation Branch**: more aggressive augmentation are used than basic augmentation. (c) **OOD Detection**: the trained model evaluates $S^{basic}$ and $S^{heavy}$ of two branch. $\tau$ is computed by $\mu(S^{basic})$ and $\sigma(S^{basic})$. If $s_i^{heavy} \in S^{heavy}$ is smaller than $\tau$, the corresponding heavy augmented data $x_i^{heavy}$ is regarded as OOD. (d) **Mix of two branch**: OOD data in heavy augmentation branch are replaced by corresponding basic augmentation version.

where $K = [\kappa_1, \kappa_2 \cdots \kappa_M]$ is the set of transformation parameters. In $\Phi(x, K)$, each individual transformation $\phi$ is defined as:

$$\tilde{x} = \phi(x, \kappa) \tag{2}$$

where $\kappa$ represents the parameter for controlling the transformation.

In this work, we propose a two-branch data augmentation framework, DualAug, as shown in Figure 3. The two branches are the basic augmentation branch and the heavy augmentation branch. The heavy augmentation branch enriches the data diversity of basic augmentations, while the basic augmented data and the training model are used to detect OOD samples from the heavy augmentation branch and replace them to corresponding basic augmentation version. In the following two subsections, we will introduce the two branches in more details.

## 3.2 THE BASIC AUGMENTATION BRANCH

We choose a baseline automated augmentation method to build the basic augmentation branch. For the convenience of description, AutoAugment (Cubuk et al., 2019) which seeks augmentation policies via reinforcement learning is taken as an example [1]. Its augmentation pipeline $\Phi(\cdot, K)$ consists of 5 sub-groups of augmentation operations, each containing two individual transformations $\phi(\cdot, \kappa)$, where $\kappa$ contains three parameters: 1) the type of transformation, 2) the probability of applying the transformation, and 3) the magnitude of the transformation. For each image, a random sub-group is selected to apply.

The basic augmentation is formulated similar to Eq. (1):

$$\tilde{x}_i^{basic} = \Phi(x_i, K^{basic}) \tag{3}$$

where $K^{basic}$ denotes the augmentation parameters in the baseline automated augmentation method.

## 3.3 THE HEAVY AUGMENTATION BRANCH

In the heavy augmentation branch, we use more aggressive augmentation parameters $K^{heavy}$ than in the basic augmentation branch. There are several different ways of realizing heavy augmentation,

---

[1] Besides AutoAugment (Cubuk et al., 2019), there are a large number of methods to be chosen from, such as RandAugment (Cubuk et al., 2020), Fast AutoAugment (Lim et al., 2019) and Deep Autoaugment (Zheng et al., 2022),*etc.*

such as increasing the number of transformations, growing the magnitude of transformations, adding more types of transformations, and enriching their combination. We implement it by increasing the number of transformations due to its convenience of being applied to automated augmentation methods.

Like the basic augmentation, heavy augmentation can be formualted as:

$$\tilde{x}_i^{heavy} = \Phi(x_i, K^{heavy}) \tag{4}$$

and we can rewrite it as follows:

$$\Phi(\cdot, K^{heavy}) = \Phi(\cdot, K^{basic}) \circ \Phi(\cdot, K^{extra}) \tag{5}$$

where $K^{basic}$ is the same as in Eq. (3) and $K^{extra}$ denotes the extra parameters. The effectiveness of different heavy augmentation implementations is discussed in Section 4.4.

### 3.4 OOD DETECTION AND THE MIX OF TWO BRANCHES

Following Hendrycks & Gimpel (2016); Liang et al. (2017), MSP (with temperature) is adopted as a score to decide whether an augmented sample $\tilde{x} \in \tilde{X}$ is in-distribution or OOD data. The model training on the fly is used for calculating the probabilities, and an apparent advantage of using it instead of a pre-trained model is the saving of memory and computation. The OOD score of $\tilde{x}_i$ can be obtained by

$$s_i = \max_{j \in \{1,...,C\}} \frac{\exp(f_\theta^j(\tilde{x}_i)/T)}{\sum_{c=1}^{C} \exp(f_\theta^c(\tilde{x}_i)/T)}, \tag{6}$$

where $f_\theta^j(\tilde{x}_i)$ represents the logit of $j$-th class and $T$ represents the temperature. Therefore, we have $s_i^{basic}$ for $\tilde{x}_i^{basic}$ and $s_i^{heavy}$ for $\tilde{x}_i^{heavy}$, respectively.

Samples with high scores are classified as in-distribution samples, and samples showing lower scores are considered as OOD. Let $S_{basic} = \{s_i^{basic}\}$ be the set which collects all scores of the augmented data from the basic branch, we know that most elements of $S_{basic}$ are in-distribution, based on which we simply assume all elements of $S_{basic}$ form an Gaussian distribution and set the threshold for filtering OOD data following the $3\sigma$ rule of thumb. That is, we use

$$\tau = \mu(S^{basic}) - \lambda \cdot \sigma(S^{basic}) \tag{7}$$

as the threshold, where $\lambda$ is a hyper-parameter.

In order not to increase the sample complexity of augmentation, we mix augmented data from the two branches, in the following manner

$$\tilde{x}_i^{dual} = \begin{cases} \tilde{x}_i^{heavy}, & if\ s_i^{heavy} > \tau \\ \tilde{x}_i^{basic}, & otherwise \end{cases} \tag{8}$$

That is, $\tilde{X}^{dual} = \{\tilde{x}_i^{dual}\}_{i=1}^{N}$ is used to train the deep network $f_\theta$.

## 4 EXPERIMENTS

In this section, we evaluate DualAug in three different tasks: supervised learning (Section 4.1), semi-supervised learning (Section 4.2), and contrastive self-supervised learning (Section 4.2). DualAug effectively enhances the performance of data augmentation in the mentioned tasks. Additionally, we provide an ablation study in Section 4.4.

### 4.1 SUPERVISED LEARNING

#### 4.1.1 EXPERIMENTAL SETTINGS

In our experiments, the augmentation parameters of DualAug include $K^{basic}$ and $K^{extra}$. $K^{basic}$ can choose from various automated data augmentation methods, such as AutoAugment (AA) (Cubuk et al., 2019), RandAugment (RA) (Cubuk et al., 2020), and Deep AutoAugment (DeepAA) (Zheng

et al., 2022). We follow the original augmentation parameters or policy of each corresponding method. The extra augmentation include $M$ extra operations, we have $K^{extra} = \{\kappa\}_{i=1}^M$. We make $M$ a random number and sample it uniformly from the range $[1, 10]$ for CIFAR-10/100 and SVHN-Core, $[1, 6]$ for ImageNet, respectively. For calculating the OOD score and threshold, the temperature $T$ and $\lambda$ are set to 1000 and 1, respectively. Since the OOD Score of the model is unreliable in the early stage of training, we will only use the basic augmentation branch at the initial training iterations (which is referred to as the warm-up stage of our DualAug). The warm-up stage is set to be the first 20% training epochs in all experiments. Other information including the training batch size, number of train epochs, and the optimizer will be introduced in appendix.

### 4.1.2 CIFAR-10/100 AND SVHN-CORE

**Results.** Table 1 reports the Top-1 test accuracy on CIFAR-10/100 (Krizhevsky et al., 2009), and SVHN-Core (Netzer et al., 2011) for WideResNet-28-10 and WideResNet-40-2 (Zagoruyko & Komodakis, 2016), respectively. When combined with automated data augmentation, DualAug demonstrates superior on all networks and datasets. It can prove that the parameters proposed by previous automated augmentation methods are tend to being conservative, which becomes an obstacle to further growth in performance. Due to the heavy augmentation variation branch and effective mix strategy, DualAug further augments data to boost model's performance.

Table 1: Top-1 test accuracy (%) on CIFAR-10, CIFAR-100 and SVHN-Core. Better results in comparison are shown in bold. Statistics are computed from three runs.

| Dataset | Metric | AA | AA+Ours | RA | RA+Ours | DeepAA | DeepAA+Ours |
|---------|--------|-----|---------|-----|---------|--------|-------------|
| CIFAR-10 | WRN-40-2 | $96.34 \pm .10$ | $\mathbf{96.44} \pm .04$ | $96.30 \pm .14$ | $\mathbf{96.48} \pm .10$ | $96.45 \pm .07$ | $\mathbf{96.48} \pm .08$ |
| | WRN-28-10 | $97.36 \pm .05$ | $\mathbf{97.44} \pm .09$ | $97.13 \pm .05$ | $\mathbf{97.32} \pm .12$ | $97.50 \pm .08$ | $\mathbf{97.59} \pm .07$ |
| CIFAR-100 | WRN-40-2 | $79.63 \pm .26$ | $\mathbf{79.83} \pm .19$ | $78.13 \pm .12$ | $\mathbf{78.57} \pm .07$ | $78.41 \pm .07$ | $\mathbf{78.47} \pm .11$ |
| | WRN-28-10 | $83.04 \pm .20$ | $\mathbf{83.42} \pm .21$ | $83.11 \pm .26$ | $\mathbf{83.60} \pm .23$ | $84.14 \pm .12$ | $\mathbf{84.39} \pm .21$ |
| SVHN-Core[*] | WRN-40-2 | $97.71 \pm .07$ | $\mathbf{98.00} \pm .09$ | $97.95 \pm .12$ | $\mathbf{98.11} \pm .03$ | - | - |
| | WRN-28-10 | $97.76 \pm .19$ | $\mathbf{98.08} \pm .11$ | $97.97 \pm .15$ | $\mathbf{98.15} \pm .06$ | - | - |

[*] DeepAA does not report its policy on SVHN-Core, thus we do not test our method with it on SVHN-Core.

### 4.1.3 IMAGENET

**Compared method.** We compare our DualAug with the following methods: 1) AA (Cubuk et al., 2019), 2) RA (Cubuk et al., 2020), 3) DeepAA (Zheng et al., 2022), 4) TeachAugment (Suzuki, 2022), 5) FastAA (Lim et al., 2019), 6) Faster AA (Hataya et al., 2020), 7) UA (LingChen et al., 2020), 8) TA (Müller & Hutter, 2021). The code of training AA and RA on ImageNet is not released, thus we reproduce them using the training settings of DeepAA.

**Evaluation protocol.** In addition to the standard ImageNet-1K test set, we have chosen to assess the effectiveness of our DualAug using more challenging datasets. These datasets include ImageNet-C (Hendrycks & Dietterich, 2019), ImageNet-C̄ (Mintun et al., 2021), ImageNet-A (Hendrycks et al., 2021), and ImageNet-V2 (Recht et al., 2019). It is noted that, to refrain from potential domain conflicts, we have eliminated contrast and brightness variations from the ImageNet-C dataset. Furthermore, we incorporate the RMS calibration error on ImageNet-1K test set as a metric to evaluate the robustness of models, following previous work (Hendrycks et al., 2019).

**Result.** In Table 2, we present the results of DualAug on the ImageNet dataset. By combining DualAug with AA, RA, and DeepAA, we observe an improvement in the performance of all three data augmentation techniques. This illustrates the effectiveness of DualAug on large-scale datasets. Specifically, DualAug with DeepAA (the state-of-the-art automated data augmentation method) achieves the highest top-1 accuracy (78.44%). In Table 3, we evaluate DualAug on more challenging ImageNet variant datasets. When combined with DualAug, automated data augmentation show improvements in all of them. This demonstrates that

Table 2: Top-1 test accuracy (%) on ImageNet.

| Method | Accuracy (%) |
|--------|-------------|
| Default | 76.40 |
| FastAA* | 77.60 |
| Faster AA* | 76.50 |
| UA* | 77.63 |
| TA* | 78.04 |
| TeachAugment* | 77.80 |
| AA | 77.30 |
| AA+Ours | 77.46 |
| RA | 76.90 |
| RA+Ours | 77.25 |
| DeepAA | 78.30 |
| DeepAA+Ours | **78.44** |

[*] Reported in previous papers.

Table 3: Result of calibration RMS on ImageNet and accuracy (%) on variants of ImageNet (including ImageNet-C, ImageNet-C̄, ImageNet-A, and ImageNet-V2) when different basic augmentations are combined with our DualAug. **Lower is better for the calibration RMS.** The result in bold is better in comparison.

| Method | Calibration RMS ($\downarrow$) | ImageNet-C | ImageNet-C̄ | ImageNet-A | ImageNet-V2 |
|---|---|---|---|---|---|
| AA | 6.77 | 38.95 | 39.67 | 5.72 | **66.00** |
| AA+Ours | **6.75** | **40.99** | **40.12** | **6.05** | 65.80 |
| RA | 7.91 | 38.99 | 38.96 | 4.09 | 65.10 |
| RA+Ours | **7.65** | **41.67** | **40.44** | **4.87** | **65.30** |
| DeepAA | 6.03 | 42.82 | 40.94 | 6.57 | 65.80 |
| DeepAA+Ours | **5.59** | **43.85** | **42.08** | **6.83** | **66.20** |

DualAug's effective mixing strategy successfully prevents excessive out-of-distribution (OOD) data from hindering the learning process. Due to its more aggressive augmentations DualAug naturally has an advantage in handling challenging tasks.

**Discussion about TeachAugment.** Similar to our DualAug in motivation, TeachAugment (Suzuki, 2022) also considers augmentation data that is meaningless or difficult to recognize in Adversarial AutoAugment (Zhang et al., 2019). As shown in Table 2, our DualAug outperforms TeachAugment which utilizes a teacher model and requires updating it with less time and memory cost. In addition, DualAug can be combined with a broader range of augmentation methods, while TeachAugment is limited to Adversarial AutoAugment only.

## 4.2 SEMI-SUPERVISED LEARNING

Semi-supervised learning (Yang et al., 2022) improves data efficiency in machine learning by utilizing both labeled data and unlabeled data. A series of semi-supervised learning methods utilize consistency training. Examples of these methods include UDA (Xie et al., 2020), FixMatch (Sohn et al., 2020), SoftMatch (Chen et al., 2023), among others. These methods constraint consistency between the outputs of two different data views, allowing the model to learn features from unlabeled data. Data augmentation plays a crucial role in helping the models learn

Table 4: Semi-supervised learning results on CIFAR-10 using WideResNet-28-2. "4000 Labels" denotes that 4,000 images have labels while the other 4,6000 do not. Similar for "250 Labels".

| Method | 250 Labels | 4000 Labels |
|---|---|---|
| FixMatch | $95.04 \pm .68$ | $95.77 \pm .06$ |
| FixMatch+Ours | $\mathbf{95.23 \pm .45}$ | $\mathbf{96.10 \pm .05}$ |

invariants of data, guided by some consistency constraints. Therefore, having high-quality data augmentation is of utmost importance in these methods.

In this section, we demonstrate the effectiveness of DualAug in semi-supervised learning by combining it with FixMatch as an example. FixMatch first generates pseudo-labels by using the model's predictions on weakly-augmented unlabeled data. A pseudo-label is only kept if the model makes a highly confident prediction for a given data. The model is then trained to predict the pseudo-label when being fed a strongly-augmented version of the same data. We apply our DualAug only to the strongly-augmented data. The experiment setting of FixMatch is presented in appendix. The setting for DualAug is the same as the one described in Section 4.1.1.

As shown in Table 4, we present the results of FixMatch with DualAug on CIFAR-10 using 250 and 4000 labelled samples only. The results show that our DualAug consistently improves the performance of FixMatch.

## 4.3 CONTRASTIVE SELF-SUPERVISED LEARNING

Contrastive learning, such as MoCo (He et al., 2020), SimCLR (Chen et al., 2020), SwAV (Caron et al., 2020), and SimSiam (Chen & He, 2021), is a crucial technique for self-supervised learning. One key step in many contrastive learning methods is to generate two different views of each single training image using data augmentations. Our DualAug significantly increases the data diversity while preserving the data distribution, thus it is capble of providing more challenging and meaningful views for contrastive learning. In this regard, we consider integrating our DualAug into contrastive learning be beneficial.

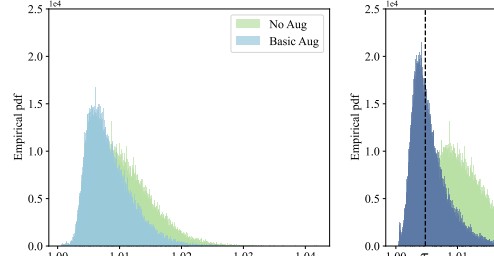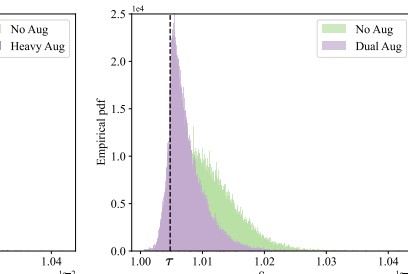

Figure 4: Score distribution of the total CIFAR-100 training set, which is obtained from the currently trained model (using WRN-40-2) at the 100-th epoch. "Empirical pdf" refers to the empirical probability density function, "Basic/Heavy/Dual Aug" refer to basic/heavy/dual augmentation, and "No Aug" refers to the original data without augmentation. $\tau$ is threshold computed by Eq (7) and averaged by iterations in one epoch. AutoAugment is chosen as basic augmentation.

We try our DualAug on SimSiam (Chen & He, 2021). We selected the original data augmentations mentioned in the SimSiam paper to serve as the basic augmentations for our DualAug, which include geometric augmentation, color augmentation, and blurring augmentation. The pre-train is conducted using ResNet-18 for 800 epochs on CIFAR-10, and ResNet-50 for 100 epochs on ImageNet, respectively. The linear classification evaluation is conducted for 100 epochs on CIFAR-10 and 90 epochs on ImageNet, respectively. We follow other experimental settings of SimSiam (which details is presented in appendix), and specific settings for DualAug are consistent with those detailed in Section 4.1.1.

As shown in Table 5, we show the linear evaluation results of pretrained SimSiam model on CIFAR-10 and

Table 5: The linear evaluation results of contrastive self-supervised learning with DualAug.

| Method | CIFAR-10 | ImageNet |
|---|---|---|
| SimSiam[1] | 91.80 | 68.10 |
| SimSiam+TeachAug[2] | - | 68.20 |
| SimSiam+RA[2] | - | 68.00 |
| SimSiam+TA[2] | - | 62.70 |
| SimSiam+AA | - | 67.85 |
| SimSiam+YOCO[3] | - | 68.30 |
| SimSiam | 91.61 | 68.23 |
| SimSiam+Ours | **92.29** | **68.67** |

[1] Reported in SimSiam.
[2] Reported in Teach Augment.
[3] Reported in YOCO.

ImageNet when integrated with DualAug. Automated data augmentation (RA, TA, and AA) has a negative impact on SimSiam. TeachAug (Suzuki, 2022) and YOCO (Han et al., 2022) enhance SimSiam's performance. DualAug consistently also improves SimSiam and outperforms previous data augmentation methods.

## 4.4 ABLATION STUDY

**Component.** As shown in Table 6, we show the ablation study to analyze the impact of different components in our DualAug. These components include the basic augmentation branch, the heavy augmentation branch, and the OOD detector. For the basic augmentation, we chose AutoAugment. Firstly, we observe a significant performance drop when only the heavy augmentation is applied (2nd row) compared to 1st row. Secondly, when we add the OOD detector without using basic augmentation (3rd row), the performance slightly improved compared to the 1st row. Thirdly, when randomly mixing the heavy augmented data and basic augmented data without the OOD detector (4th row), a slight performance drop is observed compared to the 1st row. Finally, when all three components of DualAug are used (5th row), we achieve the best performance. Moreover, compared with the 1st, 2nd, and 5th row's score distribution in Figure 4, DualAug shifts the score distribution to a larger extent while ensuring that most samples' scores are still above the threshold. This confirms that DualAug makes data sufficiently augmented and filters OOD data generated by heavy augmentation as much as possible.

Table 6: Ablation study about DualAug component using WRN-28-2 on CIFAR-100. "Basic/Heavy Aug" refers to the basic/heavy augmentation branch.

| Basic Aug | Heavy Aug | OOD Detector | Accuracy(%) |
|---|---|---|---|
| ✓ | ✗ | ✗ | 78.34 ± .10 |
| ✗ | ✓ | ✗ | 73.32 ± .25 |
| ✗ | ✓ | ✓ | 78.40 ± .19 |
| ✓ | ✓ | ✗ | 77.43 ± .24 |
| ✓ | ✓ | ✓ | **78.89** ± .21 |

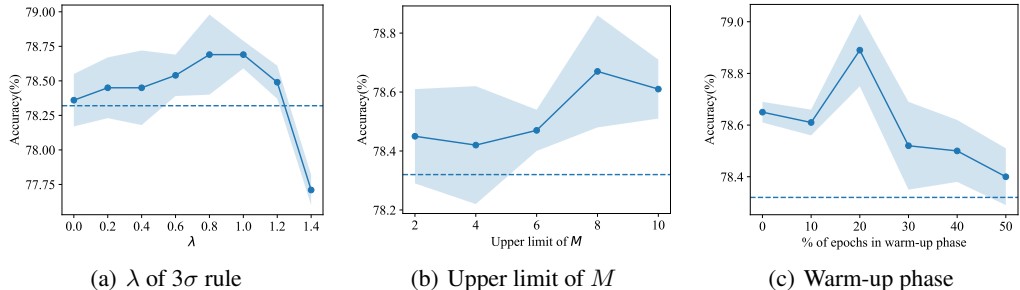

(a) $\lambda$ of $3\sigma$ rule       (b) Upper limit of $M$       (c) Warm-up phase

Figure 5: Hyper-parameters analysis of DualAug. Experiments are conducted on CIFAR-100 using WideResNet-28-2. The dashed lines represent the accuracy of basic augmentation(AutoAugment).

**OOD detector.** We also make an ablation study on OOD detector generation options. As shown in Table 7, the "Online (✓)" yields better results compared to the "Online (✗)". Furthermore, the "Online (✓)" is more efficient for time and memory saving.

**Hyper-parameter analysis.** Then, we investigate the effect of three hyper-parameter of DualAug, including $\lambda$ of $3\sigma$ rule mentioned in Section 3.4, upper limit of $M$ and the percentage of epochs in warm-up phase mentioned in Section 4.1.1. We perform experiments on CIFAR-100 dataset with WRN-28-2 backbone and evaluate on AutoAugment. According to the result shown in Figure 5(a), it is reasonable to set $\lambda$ in [0.6,1.2]

Table 7: Ablation study about OOD Detector generation using WRN-28-10 on CIFAR-100. **Online (✓)**: currently-trained model used as OOD detector. **Online (✗)** : pretrained model used as OOD detector.

| Method | Online | Accuracy(%) |
|---|---|---|
| AA | - | 83.04 ± .20 |
| AA + Ours | ✗ | 83.34 ± .15 |
| AA + Ours | ✓ | **83.42** ± .21 |

for all our experiments. According to the result shown in Figure 5(b), more aggresive upper limit of $M$ bring better performance, $M = 8/10$ is suitable. According to the result shown in Figure 5(c), 20% epochs of total stage bring the best performance.

**Heavy augmentation implementation analysis.** As mentioned in Section 3.3, there are various implementation ways to use heavy augmentation. In Table 8, RA (Cubuk et al., 2020) is chosen as an example to analyze the optimal heavy augmentation implementation manner. In detail, three different methods for heavy augmentation are chosen. Firstly, more types of transformations such as Gaussian, Blur, Sample Pairing, *etc.*are included into the transformations set in heavy augmentation. Additionally, a higher degree of transformation magnitude is used. While RA uses an magnitude of 14 in their original paper, Our heavy Augmentation uses an magnitude of 18. Finally, the number of augmentation transformations are increased, which is the DualAug's implementation. Results indicate that increasing the number of transformations is the best choice, and it can be easily applied to various basic augmentations.

Table 8: Ablation study about heavy augmentation implementation using WRN-28-10 on CIFAR-100. The word in parentheses represents implementations of heavy augmentation, which includes more *Type*, bigger *Magnitude* and extra *Number* of transformations.

| Method | Accuracy (%) |
|---|---|
| RA | 83.11 ± .26 |
| +Ours (Type) | 82.10 ± .18 |
| +Ours (Magnitude) | 83.21 ± .14 |
| +Ours (Number) | **83.60** ± .23 |

# 5 CONCLUSION

This paper revealed that a large amount of well-augmented data can still be exploited in heavy-augmentation while the weakness introduced by some OOD data outweighs the benefit of these well-augmented data. Based on this observation, we proposed a generic yet straightforward dual-branch framework, named DualAug, for automatic data augmentation. Specifically, the network is encouraged to take advantage of heavy-augmentation by data mixing strategy, which makes augmented data from the basic- and the heavy-augmentation branches wisely fuse. Extensive experiments show the effectiveness of the proposed DualAug. Moreover, the experiments of semi-supervised consistent learning and contrastive self-supervised learning prove that our DualAug can be generalized to other task settings.

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

# A    ALGORITHM

Algorithm 1 describe the pseudo code of Dual Augmentation.

---

**Algorithm 1** Dual Augmentation

---

**Input:** Dataset $D = \{(x_i, y_i)\}_{i=1}^N$, model $f_\theta$, temperature $T$ of Maximum Softmax Probability, $\lambda$ of $3\sigma$ rule, warmup period epoch $w$, mini-batch size $B$;

**Initialization:** Perform random initialization for $\theta$ of the model;

**for** each epoch **do**

  **for** each mini-batch **do**

    Sample mini-batch data $X = \{x_i\}_{i=1}^B$, $Y = \{y_i\}_{i=1}^B$ from $D$

    Augment $x \in X$ to $\tilde{x}_i^{basic} \in \tilde{X}^{basic}$ and $\tilde{x}_i^{heavy} \in \tilde{X}^{heavy}$ by Eq 3 and Eq 4

    Calculate $s_i^{basic} \in \tilde{S}^{basic}$ and $s_i^{heavy} \in \tilde{S}^{heavy}$ by Eq 6

    $\tau = \mu(S^{basic}) - \lambda \cdot \sigma(S^{basic})$

    **if** epoch $\geq w$ **then**

$$\tilde{X}^{dual} = \{\tilde{x}_i^{dual} | \begin{cases} \tilde{x}_i^{heavy}, & \text{if } s_i^{heavy} > \tau \\ \tilde{x}_i^{basic}, & \text{otherwise} \end{cases}\}$$

    **else**

      $\tilde{X}^{dual} = \tilde{X}^{basic}$

    **end if**

    Compute Loss of $\tilde{X}^{dual}$ and $Y$

    Update $\theta$

  **end for**

**end for**

**Output:** model $f_\theta$

---

# B    TRAINING SETTINGS

For all basic augmentations, our experiments use the original settings and policy of augmentation, including the range, probability, and magnitude of transformations.

## B.1    SUPERVISED LEARNING

For CIFAR-10/100 and SVHN-Core the WideResNet-40-2 and WideResNet-28-10  (Zagoruyko & Komodakis, 2016) network is trained for 200 epochs using SGD with Nesterov Momentum, a learning rate of 0.1, a batch size of 128, a weight decay of 5e-4, and cosine learning rate decay.

For ImageNet (Deng et al., 2009), the ResNet-50 (He et al., 2016) is trained using the training setup of DeepAA (Zheng et al., 2022). The training process lasts for 270 epochs, with each GPU using a batch size of 512. The training is conducted on 2 A6000 GPUs, using image crops of size 224 x 224. The initial learning rate is set to 0.1. A stepwise 10-fold reduction is applied after 90, 180, and 240 epochs. A linear warmup factor of 8 over the first 5 epochs is used.

## B.2    SEMI-SUPERVISED LEARNING

We train the FixMatch for 1024 epochs on a 2080Ti GPU using a batch size of 64. The initial learning rate is set to 0.03, and we adopt a cosine learning rate schedule. The weight decay is set at $5e-4$. The coefficient for the unlabeled batch size is 7, the coefficient for the unlabeled loss is 1, the pseudo-label temperature is 1, and the pseudo-label threshold is 0.95.

## B.3    CONTRASTIVE SEMI-SUPERVISED LEARNING

For CIFAR-10, we pre-train SimSiam on a single 3060Ti GPU using a batch size of 512 for 800 epochs. The learning rate adjusts using cosine scheduling, starting at 0.05, with a weight decay of $5e-4$. For the linear evaluation of SimSiam, it runs for 100 epochs with a batch size of 256. The initial learning rate is set to 30 and decays by a factor of 0.1 at the 60th and 80th epochs. The momentum is set at 0.9.

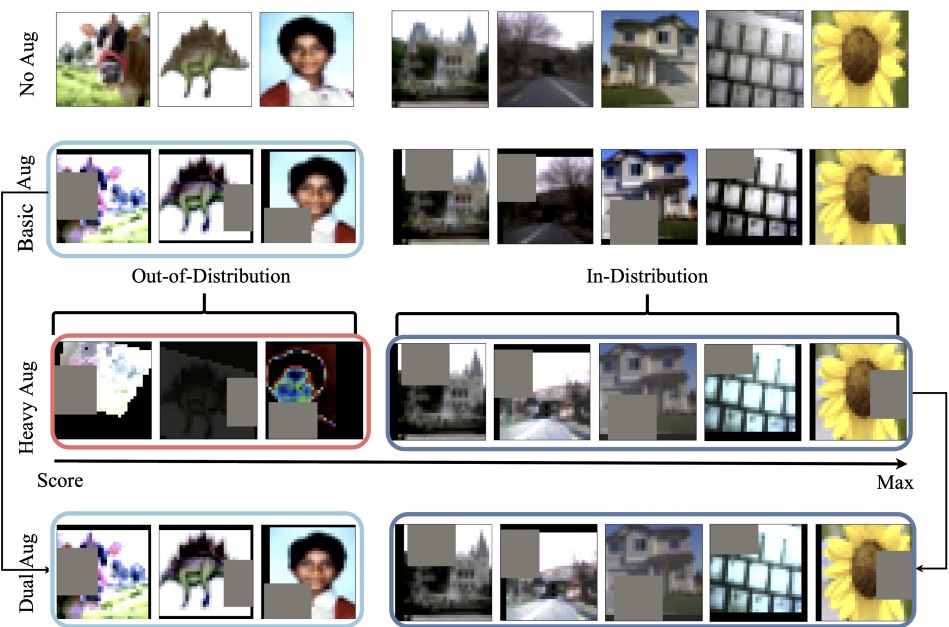

Figure 6: Visualization on CIFAR-100 of the original image, basic augmented image, heavy augmented image, and dual augmented image. Basic/Heavy/Dual Aug refer to basic/heavy/Dual augmentation, and No Aug refers to the original data without augmentation. The Heavy Aug row is sorted from min to max based on the image's score, with corresponding images in the other rows.

For ImageNet, we pre-train SimSiam on two A6000 GPUs with a batch size of 512. The learning rate follows a cosine adjustment, starting from 0.05, with a weight decay of $5e - 4$. During the linear evaluation of SimSiam, which lasts 100 epochs with a batch size of 256, the initial learning rate is set to 0.1 and employs a cosine learning rate schedule. The momentum is set at 0.9.

## C   DUALAUG WITH FASTAUTOAUGMENT AND UNIFORMAUGMENT

As a supplement, the results of combining DualAug with FastAutoAugment (Lim et al., 2019) and UniformAugment (LingChen et al., 2020) are presented in Table 17. It can be seen that DualAug effectively improves the performance of basic augmentation, except for the comparable performance of WideResNet-40-2 on CIFAR-100 for FastAutoAugment.

Table 9: Result of DualAug with FasetAA and UA. Result in bold is better in comparison.

| Dataset | Model | FastAA | | UA | |
|---|---|---|---|---|---|
| | | Ours (✗) | Ours (✓) | Ours (✗) | Ours (✓) |
| CIFAR-10 | WRN-40-2 | 96.40 | **96.65** | 96.25 | **96.53** |
| | WRN-28-10 | 97.30 | **97.54** | 97.33 | **97.58** |
| CIFAR-100 | WRN-40-2 | 79.40 | **79.40** | 79.01 | **79.74** |
| | WRN-28-10 | 82.70 | **83.16** | 82.82 | **83.34** |

## D   VISUALIZATION OF DUAL AUGMENTED IMAGE

Figure 6 shows the visualization on CIFAR-100 of the original image, image of basic augmentation, heavy augmentation, and dual augmentation. Among them, the images in the row of Heavy Aug are sorted from minimum to maximum according to the Score of the image, and the other rows are the corresponding images of the Heavy Aug row. As described in Section 3.4, heavy augmented images that is out-of-distribution is degraded to basic augmented images, and dual augmented images mix heavy augmented images and basic augmented images.

## E   DUALAUG'S FAIR COMPARISON WITH TEACHAUGMENT AND KDFORAA

**KDforAA.** We present a fair comparison between DualAug and KDforAA (Wei et al., 2020), as shown in Table 10. The experiments are based on AutoAugment and use the WRN-28-10 model. To ensure fairness, the settings and Memory/Time cost are aligned with those of KDforAA. DualAug demonstrate better performance compared to KDforAA.

**TeachAugment.** We provide a fair comparison between TeachAugment (Suzuki, 2022) and DualAug in the same training pipeline, as shown in Table 11. The experiments use the WRN-28-10 model. To ensure fairness, we use the EMA model to filter OOD data in Adv. Aug+DualAug, as done in TeachAugment. TeachAug and DualAug present comparable performance in the fair setting, using the Adv. Aug (Zhang et al., 2019) as the baseline. Furthermore, compared to TeachAug, we are able to improve upon a more widely automated augmentation method(DeepAA+DualAug), resulting in better performance in the same training pipeline. It is also worth noting that DualAug exhibits better performance in comparison to TeachAug in self-supervised learning task, as shown in Table 5 (68.67% vs. 68.20%).

Table 10: Comparison between KDforAA and Ours. Result in bold is the best.

| Dataset | KDforAA* | AA + Ours |
|---------|----------|-----------|
| CIFAR-10 | 97.6 | **97.63** ± .09 |
| CIFAR-100 | 83.8 | **83.94** ± .26 |

\* Reported in KDforAA

Table 11: Comparison between TeachAugment and Ours. Result in bold is the best.

| Dataset | TeachAug | Adv. Aug + Ours | DeepAA+Ours |
|---------|----------|-----------------|-------------|
| CIFAR-10 | 97.25 ± .11 | 97.35 ± .09 | **97.52** ± .18 |
| CIFAR-100 | 83.08 ± .39 | 83.02 ± .34 | **84.54** ± .23 |

## F COMPUTATIONAL COST OF DUALAUG AND OTHER METHODS

We present a comparison of computation costs in Table 12. The experiment is carried out on CIFAR-100 using a single 2080Ti GPU with WRN-28-10 model. As shown, when considering memory cost, DualAug use the same memory cost as AA [2], while TeachAug and KDforAA require additional memory. When it comes to time cost, DualAug use the least additional time cost. [3] All in all, considering all factors, DualAug is a reasonable choice for achieving better performance in comparison to KDforAA and TeachAug.

## G DUALAUG'S RESULTS ON THE FGVC DATASET

We present more results on Flowers(Nilsback & Zisserman, 2008), Caltech(Fei-Fei et al., 2004), Pets(Em et al., 2017), Aircraft(Maji et al., 2013), Cars(Krause et al., 2013) in Table 13. The experiments train the inception-v4 model for 1000 epoch. DualAug demonstrates a significant improvement over AutoAugment in almost all FGVC datasets except for Pets.

Table 12: The computational cost of DualAug and other methods

| Method | Time Cost | Memory Cost |
|--------|-----------|-------------|
| AA | 5.6h | 4453MiB |
| AA+Ours | **8.6h** | **4453MiB** |
| KDforAA | 12.9h | 5804MiB |
| TeachAug | 9.5h | 8714MiB |

Table 13: Results of DualAug on FGVC dataset. Result in bold is better in comparison.

| Method | Flowers | Caltech | Pets | Aircraft | Cars | Avg. |
|--------|---------|---------|------|----------|------|------|
| AA | 87.55% | 81.76% | **84.52%** | 85.63% | 93.51% | 86.59% |
| AA+Ours | **88.92%** | **84.85%** | 83.70% | **86.74%** | **94.01%** | **87.64%** |

## H DUALAUG'S RESULTS ON THE OBJECT DETECTION TASK

DualAug can also be extended to other tasks, such as object detection. We can analyze its effectiveness in two phases: the pre-training phase and the finetuning phase.

---

[2]In our implementation, DualAug computes the scores of two branches one after another, thus it does not incur any additional memory cost.

[3]The time cost of KDforAA includes the time spent training the teacher model.

In the pre-training phase. Our pre-training backbone can also be leveraged for object detection and other related tasks. When we expand the results of Table 5 to the object detection task, we find that DualAug leads to improvements in performance in Table 14.

In the finetuning phase, when DualAug is applied to a specific task, its softmax score can also be utilized for out-of-distribution (OOD) detection. Following Zoph et al. (2020), we conduct an experiment to demonstrate the performance of DualAug in object detection in Table 15.

Table 14: Results of classification and object detection on the pre-train Simsiam model.

| Method | Classification(Acc. %) | VOC 07+12 Detection(AP) |
|---|---|---|
| Simsiam | 68.20 | 51.67 |
| Simsiam+Ours | **68.67** | **52.02** |

Table 15: Results of DualAug on object detection finetune phase.

| Method | VOC 07 Detection(AP) |
|---|---|
| AA | 56.30 |
| AA+Ours | **56.73** |

Expanding DualAug to encompass a wider range of tasks is an exciting aspect of future work. We believe it will lead to further exploration and advancement in the field.

## I  DISCUSSION ABOUT DUALAUG'S ADDITIONAL COMPUTATIONAL COST

We conduct experiments to ensure that the training costs are aligned. We increase RA's training time to ensure a fair comparison, calling it RA*. The results in Table 16 clearly demonstrate that RA+Ours outperforms the RA* and RA. The experiment is conducted using WRN-28-2 on CIFAR-100.

It is important to note that the policy search cost for RA, AA, and other automated augmentation methods is quite expensive, despite not incurring extra costs in training (as reported in the table below and DeepAA (Zheng et al., 2022)'s Table 4). Therefore, the increase in computational cost is slight and reasonable when compared to the significant performance improvement that DualAug provides.

Table 16: Analysis of DualAug's additional computational cost

| Method | Accuracy(%) | Search Time(h) | Train Time(h) |
|---|---|---|---|
| RA | 77.94±.15 | 25 | 1 |
| RA* | 78.11±.13 | 25 | 1.5 |
| RA+Ours | **78.46**±.12 | 0 | 1.5 |

## J  AN ABLATION STUDY OF DUALAUG'S OOD SCORE

From a theoretical perspective, it appears that refining the metrics for the OOD score could improve the performance of DualAug. However, when integrating it into DualAug, the trade-off between computational expenses of OOD score calculation and the gain of performance must be carefully weighed within the context of the specific task. We attempt to use energy-base model(EBM) (Liu et al., 2020) as an alternative OOD Score. EBM may have a slight edge over softmax. The experiment is conducted using WRN-40-2 on CIFAR-100.

Table 17: Results of DualAug using different OOD score

| Method | Accuracy(%) |
|---|---|
| AA+Ours(SoftMax) | 79.83 ± .19 |
| AA+Ours(EBM) | 79.95 ± .21 |

## K  DUALAUG'S RESULTS BASED ON MIXUP

We use Mixup (Zhang et al., 2017) as an example to analyze the integration of DualAug with mixed-base augmentation. The experiment is conducted using ResNet-18 on CIFAR-10. Upon

Table 18: Results of DualAug based on Mixup

| Method | Accuracy(%) |
|---|---|
| Mixup($\alpha = 1.0$) | 95.64 ± .11 |
| Mixup($\alpha = 10.0$) | 94.67 ± .14 |
| Mixup($\alpha = 10.0$)+Ours | **95.78** ± .09 |

observation, it appears that mixup tends to select $\alpha <= 1$ in $Beta(\alpha, \alpha)$ which may not provide a enough amplitude of augmentation. To address this, we can simply set $\alpha > 1$ and utilize DualAug to filter OOD data in mixup data. The Table 18 demonstrates that we achieve some improvement compared to standard mixup.

