# OpenReview forum: "DualAug: Exploiting Additional Heavy Augmentation with OOD Data Rejection"
_ICLR.cc/2024/Conference — Submitted to ICLR 2024_

### Official Review · Reviewer_c5kH · 2023-10-12

**Soundness:** 3 good
**Presentation:** 3 good
**Contribution:** 3 good
**Rating:** 6
**Confidence:** 4

**Summary:**

This study reveals a noteworthy observation: a substantial increase in the application of data augmentation transformations for classification tasks leads to a precipitous decline in performance as shown in Figure 1. Building upon this revelation, the authors introduce an Out-of-Distribution (OOD) discarding techniques that not only safeguards performance from the unusual samples but also enhances it. Notably, this novel method harnesses two distinct branches of augmentation, namely the basic augmentation branch and the heavy augmentation branch. The acceptability of a sample is gauged based on the softmax values obtained from each branch's outcome. The authors demonstrate that this innovative approach can be seamlessly integrated into existing AutoAugmentation methodologies, resulting in performance enhancement. Furthermore, they substantiate the efficacy and superiority of their proposed technique through self-supervised performance evaluation which strongly utilize augmentation in their training procedure.

I will subsequently provide a summary of strengths and concerns from my perspective, along with any questions I have.

**Strengths:**

Strengths:
1. This paper presents a compelling insight into the AutoAugmenttion methods. It highlights a critical aspect - the profound influence of the number of augmentations on performance. Previous AutoAugmentation approaches have predominantly relied on manually selecting the number of transformations, underscoring the importance of examining this factor.

2. Building on this discovery, the proposed method stands out for its elegant simplicity and remarkable effectiveness. It employs a straightforward scoring mechanism, computed through a simple feed-forward process on the samples. In contrast, previous AutoAugmentation methods have incurred substantial computational costs in the quest for optimal augmentations. This novel approach streamline the process, utilising just one additional branch in the feed-forward stage while achieving impressive results.

3. The paper demonstrates consistent performance improvements across various scenarios. The authors showcase enhanced performance on divers datasets. Despite the extensive history of AutoAugmentation techniques, these improvements are noteworthy, as they consistently enhance results across different use cases.

**Weaknesses:**

Weaknesses:

1. While this paper demonstrates performance improvements, its applicability is primarily limited to classification tasks, including contrastive learning. As previously mentioned in the strengths section, given the extensive history of AutoAugmentation research, the room for improvement in these specific domains appears limited. Therefore, it is essential to explore the impact of this research on other datasets. For instance, leveraging additional datasets like those used in the AutoAugmentation [1] paper (Flowers, Caltech, Pets, Aircraft, Cars) could provide further insights. Moreover, the proposed method, relying on softmax scores for out-of-distribution sample detection, may require refinement when applied to other tasks, such as object detection (as DADA [2] did).


2. Despite the paper's assertion that the proposed algorithm incurs minimal computational costs by utilizing an additional branch for feed forwarding to obtain heavy and basic augmentation datasets, the authors do not provide an analysis of the algorithm's additional computational expenditure. For instance, it would be valuable to compare this algorithm's cost-effectiveness with that of existing methods like RA, which do not incur extra costs. Such an analysis could strengthen the argument for the proposed algorithm's efficacy.


3. It appears that the model in this study was trained on datasets twice as large as those used in previous algorithms, given that it utilizes data from two branches. This discrepancy could introduce an element of unfairness when comparing the proposed algorithm with its predecessors. To ensure a fair comparison, one potential approach could involve restricting the number of samples the model encounters during the training process.

4. Minor Weakness:
(Lack of reference to related work) The paper does not reference two relevant works: DADA [2], which focuses on Data Augmentation using Differentiability, and CUDA [3], which provides an analysis of the number of augmentation operations in both class imbalance and balanced tasks.

[1] AutoAugment: Learning Augmentation Strategies from Data, CVPR 2019
[2] DADA: Differentiable Automatic Data Augmentation, ECCV 2020
[3] CUDA: Curriculum of Data Augmentation for Long-tailed Recognition, ICLR 2023

**Questions:**

Here are some brief questions regarding this paper:
(1) Is it possible to explore alternative metrics for the OOD score, such as the Mahalanobis distance [4]? While the Softmax-based approach is effective, there may be even better scoring methods worth considering.


(2) Could the authors extend their analysis to encompass other tasks, such as object detection?


(3) Is there potential for the proposed method to be integrated with mixed sample data augmentation techniques like MixUp [5], CutMix [6], or similar approaches?

[4] A Simple Unified Framework for Detecting Out-of-Distribution Samples and Adversarial Attacks, NeurIPS 2018
[5] mixup: Beyond Empirical Risk Minimization, ICLR 2017
[6] CutMix: Regularization Strategy to Train Strong Classifiers with Localizable Features, ICCV 2019.

**Details Of Ethics Concerns:**

I have no ethics concerns of this paper.


-------------------------
After reviewing the authors' response, I have chosen to increase my rating from 5 to 6. I appreciate the thoroughness of your responses.

---

> ### Author Response · Authors · 2023-11-22
> **Response to Reviewer c5kH (1/7)**
>
> Thank you for your constructive and insightful reviews. The following is our response.
>
> **Weakness1: More dataset results of DualAug **
>
> We present more results on Flowers, Caltech, Pets, Aircraft, Cars as below tables. The experiment uses the inception-v4 model. DualAug demonstrates a significant improvement over AutoAugment in almost all FGVC datasets except for Pets.
> | Method | Flowers  | Caltech | Pets | Aircraft | Cars | Avg. |
> | :---: | :---: | :---: | :---: | :---: | :---: | :---: |
> | AA | 87.55% | 81.76% | **84.52%** | 85.63% | 93.51% | 86.59% |
> | AA+DualAug | **88.92%** | **84.85%** | 83.70% | **86.74%** | **94.01%** | **87.64%** |

---

> ### Author Response · Authors · 2023-11-22
> **Response to Reviewer c5kH (2/7)**
>
> **Weakness 1 & Question 2: Other tasks of DualAug**
>
> DualAug can also be extended to other tasks, such as object detection. We can analyze its effectiveness in two phases: the pre-training phase and the finetuning phase.
> In the pre-training phase, we have observed that DADA has been used to improve the ImageNet pre-training backbone for object detection. Our pre-training backbone can also be leveraged for object detection and other related tasks. When we expand the results of Table 5 to the object detection task, we find that DualAug leads to improvements in performance.
>
> | Method | Classification(Acc. %) | VOC 07+12 Detection(AP) |
> | --- | --- | --- |
> | Simsiam | 68.20% | 51.67 |
> | Simsiam+DualAug | **68.67%** | **52.02** |
>
> In the finetuning phase, when DualAug is applied to a specific task, its softmax score can also be utilized for OOD detection. Following [1], we conduct an experiment to demonstrate the performance of DualAug in object detection.
>
> | Method | VOC 07 Detection(AP) |
> | :---: | :---: |
> | AA[1] | 56.30 |
> | AA[1]+DualAug | **56.73** |
>
> Expanding DualAug to encompass a wider range of tasks is an exciting aspect of future work. We believe it will lead to further exploration and advancement in the field.
>
> [1] Learning Data Augmentation Strategies for Object Detection ECCV 2020

---

> ### Author Response · Authors · 2023-11-22
> **Response to Reviewer c5kH (3/7)**
>
> **Weakness 2: Additional computational expenditure.**
>
> We conduct experiments to ensure that the training costs are aligned. We increase RA's training time to ensure a fair comparison, calling it RA*. The results in the table below clearly demonstrate that RA+DualAug outperforms the RA* and RA. The experiment is conducted using WRN-28-2 on CIFAR-100.
>
> It is important to note that the policy search cost for RA, AA, and other automated augmentation methods is quite expensive, despite not incurring extra costs in training (as reported in the table below and DeepAA's Table 4). Therefore, the increase in computational cost is slight and reasonable when compared to the significant performance improvement that DualAug provides.
>
> | Method | Accuracy(%) | Search Time(h) | Train Time(h) |
> | --- | :---: | :---: | :---: |
> | RA | 77.94±.15 | 25 | 1 |
> | RA* | 78.11±.13 | 25 | 1.5 |
> | RA+DualAug | **78.46±.12** | 0 | 1.5 |

---

> ### Author Response · Authors · 2023-11-22
> **Response to Reviewer c5kH (4/7)**
>
> **Weakness 3: Unfair comparison with previous methods about dataset size.**
>
> It appears there may be a misunderstanding. We clarify that DualAug is using the **same large** dataset as the previous method. As shown in Figure 3 and Equation (8) of the main paper, the mix of two branch is the same size as the basic augment branch, ensuring a fair comparison.

---

> ### Author Response · Authors · 2023-11-22
> **Response to Reviewer c5kH (5/7)**
>
> **Weakness 4: Lack of reference to related work**
>
> We apologize for the negligence. We have now added the DADA and CUDA to references in the revision.

---

> ### Author Response · Authors · 2023-11-22
> **Response to Reviewer c5kH (6/7)**
>
> **Question 1: Explore alternative metrics for the OOD score.**
>
> From a theoretical perspective, it appears that refining the metrics for the OOD score could improve the performance of DualAug. However, when integrating it into DualAug, the trade-off between computational expenses of OOD score calculation and the gain of performance must be carefully weighed within the context of the specific task.
>
> Considering Mahalanobis distance in [1] is quite complex. We attempt to use energy-base model (EBS)[2] as an alternative OOD Score. EBM may have a slight edge over softmax. The experiment is conducted using WRN-40-2 on CIFAR-100.
>
> | Method | Accuracy(%) |
> | --- | :---: |
> | AA+DualAug(SoftMax) | 79.83±.19 |
> | AA+DualAug(EBM) | 79.95±.21 |
>
> [1] A Simple Unified Framework for Detecting Out-of-Distribution Samples and Adversarial Attacks, NeurIPS 2018
>
> [2] Energy-based Out-of-distribution Detection, NeurIPS 2020

---

> ### Author Response · Authors · 2023-11-22
> **Response to Reviewer c5kH (7/7)**
>
> **Question 3: Integrate DualAug with mixed sample data augmentation techniques**
>
> We use Mixup as an example to analyze the integration of DualAug with mixed-base augmentation. The experiment is conducted using ResNet-18 on CIFAR-10. Upon observation, it appears that mixup tends to select $\alpha<=1$ in $Beta(α, α)$  which may not provide an enough amplitude of augmentation. To address this, we can simply set $\alpha>1$ and utilize DualAug to filter OOD data in mixup data. The table below demonstrates that we achieve some improvement compared to standard mixup.
>
> | Method | Accuracy(%) |
> | --- | :---: |
> | Mixup($\alpha=1.0$) | 95.64±.11 |
> | Mixup($\alpha=10.0$) | 94.67±.14 |
> | Mixup($\alpha=10.0$)+DualAug | **95.78±.09** |
>
> We would like to extend our sincere appreciation once again for your valuable reviews, which have greatly improved the quality of our manuscript. We look forward to your continued discussions.

---

> ### Author Response · Authors · 2023-11-23
> **Thanks for your positive feedback**
>
> Dear Reviewer c5kH,
>
> Greatly appreciate your feedback confirming that our response met your concerns. Have a wonderful day ahead!
>
> Best regards,
>
> Authors

---

### Official Review · Reviewer_hEeP · 2023-10-31

**Soundness:** 3 good
**Presentation:** 3 good
**Contribution:** 2 fair
**Rating:** 6
**Confidence:** 4

**Summary:**

The paper presents an approach to data augmentation in deep learning, aimed at improving model performance while addressing the issue of overfitting. The authors emphasize the trade-off between data diversity and the potential degradation of data quality that can occur with heavy data augmentation. They introduce a two-branch data augmentation framework called DualAug, which aims to keep augmented data within the desired distribution. The framework consists of a basic data augmentation branch and a heavy augmentation branch, with a mechanism for detecting and filtering out-of-distribution (OOD) data.
The contributions of the paper are clearly outlined, including the identification of informative augmented data even with heavy augmentation, the importance of filtering OOD data, and the introduction of DualAug as a practical solution. The experimental results on image classification benchmarks demonstrate the effectiveness of DualAug in improving various data augmentation methods.
The related work section provides a comprehensive overview of automated data augmentation and out-of-distribution detection, highlighting the unique aspects of DualAug in comparison to existing approaches. The paper also extends its investigation to semi-supervised learning and contrastive self-supervised learning, showing that DualAug can enhance the performance of these methods. The experimental results are presented clearly and support the paper's claims.
In conclusion, this paper introduces a decent contribution to the field of data augmentation, offering a well-motivated solution to the challenges of heavy augmentation and OOD data. The paper is well-written and presents its findings effectively.

**Strengths:**

The paper has the following strengths:
* DualAug Framework: One of the primary strengths of this work is the introduction of the DualAug framework. Unlike many previous data augmentation methods that often strike a balance between data diversity and the preservation of semantic context, DualAug explicitly focuses on the problem of out-of-distribution (OOD) data caused by heavy augmentation. This framework features two branches: a basic data augmentation branch and a heavy augmentation branch, each tailored to their specific needs. This innovative approach is a fresh take on addressing the challenges associated with heavy data augmentation.
* OOD Detection Integration: The incorporation of out-of-distribution (OOD) detection within the data augmentation process is a new contribution. While previous works in data augmentation have primarily focused on generating diverse training data, this paper recognizes the importance of detecting and mitigating OOD data, which can adversely impact model performance. This integration sets the work apart from previous methods that often overlook OOD data issues.
* Integration with Existing Methods: The paper not only introduces a new approach but also demonstrates how DualAug can be integrated with existing data augmentation methods. This approach allows researchers and practitioners to benefit from the proposed framework without needing to reinvent their entire data augmentation pipelines.
* Avoidance of Additional Models: Unlike some existing methods that rely on additional models or complex optimization strategies, DualAug focuses on the simplicity of implementation. It effectively addresses unexpected augmented data without introducing extra complexity. This simplicity is an attractive feature for practitioners who seek efficient and straightforward solutions.

**Weaknesses:**

There are some potential weaknesses in the work:
* Threshold Selection for OOD Detection: The paper utilizes the 3σ rule of thumb to set the threshold for filtering out OOD data. While this is a straightforward approach, it may not be the most optimal one in practice. The choice of threshold values in OOD detection can be crucial, and it's unclear whether the 3σ rule is universally suitable for different datasets and models. A more robust method for threshold selection would enhance the reliability of the approach.
* Marginal improvement over existing Methods: One aspect of the paper that requires attention is the relatively marginal improvement in performance demonstrated by the DualAug framework, especially when compared to previous works in data augmentation. While the paper presents itself as an effective approach to addressing the challenges of data augmentation, the magnitude of the performance gains achieved is not particularly substantial.

**Questions:**

The idea is simple and clear which I like. However, the effectiveness is under question because the gains are very marginal. Hence making me question the value of this work to the research community.

---

> ### Author Response · Authors · 2023-11-22
> **Response to Reviewer hEep (1/2)**
>
> Thank you for your constructive and insightful reviews. The following is our response.
>
> **Weakness 1: Threshold Selection for OOD Detection**
>
> In almost all tasks, datasets, and models, we have observed that the 3-sigma rule **consistently yields improvements**, and we do not deliberately adjust  λ.
>
> In addition, it is a common practice to set a threshold for identifying ID/OOD samples manually in the field of OOD detection, as noted in [1]. The 3-sigma rule offers an adaptive method to dynamically adjust this threshold.
>
> [1] A Baseline for Detecting Misclassified and Out-of-Distribution Examples in Neural Networks ICLR 2017

---

> ### Author Response · Authors · 2023-11-22
> **Response to Reviewer hEep (2/2)**
>
> ** Weakness 2 & Question: Marginal improvement over existing Methods **
>
> The table below illustrates the performance of the main data augmentation methods on CIFAR-100, WRN-28-10 in recent years. The improvement achieved by DualAug is reasonable, compared to the previous methods.
> | Method | Accuracy(%) | $\Delta$ |
> | :---: | :---: | :---: |
> | AutoAugment (2018) | 82.90 | - |
> | RandAugment (2019) | 83.30 | 0.40 |
> | TrivialAugment (2021) | 83.54 | 0.24 |
> | DeepAA (2022) | 84.02 | 0.48 |
> | DeepAA+DualAug (2023) | **84.39** | 0.37 |
>
>
> Moreover, DualAug's **stable performance gain** is quite noticeable as observed by the reviewer. DualAug provides consistent improvement in different datasets, various baseline data augmentations (AA, RA, and DeepAA), and tasks (classification, semi-supervised learning, and self-supervised learning). Our extra experiment on more datasets in Response to Reviewer c5kH (1/7) also proves that DualAug is competitive.
>
> Thanks once more for the valuable feedback and insightful review you provided. I anticipate engaging in further discussions with you.

---

> ### Comment · Reviewer_hEeP · 2023-11-23
>
> Thank you authors for their response. The rebuttal seems to be answer my concerns and I would like to adjust my score by one point.

---

> > ### Author Response · Authors · 2023-11-23
> > **Thanks for your positive feedback**
> >
> > Dear Reviewer hEeP,
> >
> > Thanks very much for letting us know that our rebuttal address your concerns! Have a nice day.
> >
> > Best regards,
> > Authors

---

### Official Review · Reviewer_WegG · 2023-11-01

**Soundness:** 3 good
**Presentation:** 3 good
**Contribution:** 2 fair
**Rating:** 5
**Confidence:** 4

**Summary:**

This paper presents an approach to improve on any data augmentation method based on pre-defined transformations as those proposed in Autoaugment (Cubuk et al. 2019). The idea is to apply two different kind of transformations to the same image. The first (basic) is the original of a given method such as Autoaugment or Randaugment or Deepaugment. The second (heavy) is the basic combined with an extra one, which correspond to more M, the number of applied transformations (authors tested also more magnitudes and more types of transformations but M seems to be the most important, see table 8).
Finally an out of distribution detector based on the scores of the classification model will choose whether to apply the basic transformation or the heavy. This approach seems to provide improved results in most of the cases.

**Strengths:**

- The method is simple, can be applied to different augmentation methods and does not require extra components.
- The evaluation is performed on the most important and common datasets for image classification such as CIFAR and ImageNet.
- The method seems to improve also FixMatch a semi-supervised approach based on data augmentation and SiamSiam a self-supervised learning approach.
- Ablation studies help to characterise the method.

**Weaknesses:**

- The evaluation seems missing some important evaluations on similar approaches:
a) it is missing a comparison with [1], which seems very similar in aim and has also results on the same datasets. From table 2 in [1], results are actually a bit better in [1], which is a paper form 2020.
b) the proposed method is quite similar to TeachAugment, as reported by the authors, but the comparison is performed only on ImageNet (table 2) and without the same training. The improvement could be due to a better training pipeline. The authors should train TeachAugment on their pipeline, making sure that there are no differences in the training other than the method.
- The method seems to have quite some hyper-parameters such as \labda, M, warm-up phase, which makes it more complex for a real deployment.
- There is no clear comparison with other methods in terms of computation cost. For instance, in my understanding this approach has a higher computational complexity as Teachaugment, as it requires to evaluate both basic and heavy transformation for each sample.
- The improvement provided by the approach, although stable on different datasets, seems marginal.


[1] Wei, Longhui, et al. "Circumventing outliers of autoaugment with knowledge distillation." European Conference on Computer Vision. Cham: Springer International Publishing, 2020.

**Questions:**

- I would like to see a fair comparison with [1] and TeachAugment on the commonly used datasets.
- Could you compare the computational cost of the proposed approach and competing methods?
- How often are mean and std of the basic distribution estimated?

---

> ### Author Response · Authors · 2023-11-22
> **Response to Reviewer WegG (1/5)**
>
> Thank you for your constructive and insightful reviews. The following is our response.
>
> **Weakness 1 & Question 1: Comparison and Discussion with TeachAugment and KDforAA[1].**
>
> **KDforAA[1]**: We present a fair comparison between DualAug and KDforAA[1], as shown in the table below. The experiments are based on AutoAugment and use the WRN-28-10 model. To ensure fairness, the settings and Memory/Time cost are aligned with those of KDforAA[1]. DualAug demonstrates better performance compared to KDforAA[1].
> | Dataset  | KDforAA* | AA + DualAug |
> | :---: | :---: | :---: |
> | CIFAR-10 | 97.6 | **97.63±.09** |
> | CIFAR-100 | 83.8 | **83.94±.26** |
>
> \* : Reported in KDforAA[1].
>
> [1] Wei, Longhui, et al. "Circumventing outliers of autoaugment with knowledge distillation." ECCV, 2020.
>
> **TeachAugment**: We provide a fair comparison between TeachAugment and DualAug in the same training pipeline, as shown in the table below. The experiments use the WRN-28-10 model. To ensure fairness, we use the EMA model to filter OOD data in Adv. Aug+DualAug, as done in TeachAugment.
>
> | Dataset | TeachAug | Adv. Aug+DualAug | DeepAA+DualAug |
> | :---: | :---: | :---: | :---: |
> | CIFAR-10 | 97.25%±.11% | 97.35%±.09% | **97.52%±.18%** |
> | CIFAR-100 | 83.08%±.39% | 83.02%±.34% | **84.54%±.23%** |
>
> TeachAug and DualAug present comparable performance in the fair setting, using the Adv. Aug as the baseline. Furthermore, compared to TeachAug, we are able to improve upon a more widely automated augmentation method(DeepAA+DualAug), resulting in better performance in the same training pipeline. It is also worth noting that DualAug exhibits better performance in comparedison to TeachAug in self-supervised learning task, as shown in Table 5 (68.67% vs. 68.20%).

---

> ### Author Response · Authors · 2023-11-22
> **Response to Reviewer WegG (2/5)**
>
> **Weakness 2: Hyper-parameters of DualAug**
>
> The hyper-parameters of DualAug have shown **consistent performance** across a wide range of experiments. This includes λ for the 3σ rule, the upper limit of M, and the warm-up phase. In Sections 4.1 Supervised Learning, 4.2 Semi-Supervised Learning, and 4.3 Contrastive Self-Supervised  Learning, these parameters are **consistently** set at 1.0, 10, and 20%, with the exception of ImageNet classification. Additionally, Figure 5 illustrates their consistent performance across different settings.

---

> ### Author Response · Authors · 2023-11-22
> **Response to Reviewer WegG (3/5)**
>
> **Weakness 3 & Question 2: Computational cost of the proposed approach and other methods.**
>
> We present a comparison of computation costs in the table below. The experiment is carried out on CIFAR-100 using a single 2080Ti GPU with WRN-28-10 model. As shown, when considering memory cost, DualAug uses the same memory cost as AA, while TeachAug and KDforAA require additional memory. When it comes to time cost, DualAug uses the least additional time cost. All in all, considering all factors, DualAug is a reasonable choice for achieving better performance in comparison to KDforAA and TeachAug.
>
> | Method | Time Cost | Memory Cost  |
> | :---: | :---: | :---: |
> | AA | 5.6h | 4453MiB |
> | AA+DualAug | 8.6h | 4453MiB$^1$  |
> | KDforAA[1] | 12.9h$^2$ | 5804MiB |
> | TeachAug | 9.5h | 8714MiB |
>
> 1.  In our implementation, DualAug computes the scores of two branches one after another, thus it does not incur any additional memory cost.
> 2.  Containing the time of training the teacher model

---

> ### Author Response · Authors · 2023-11-22
> **Response to Reviewer WegG (4/5)**
>
> **Weakness 4: Improvement of DualAug**
>
> The table below illustrates the performance of the main data augmentation methods on CIFAR-100, WRN-28-10 in recent years. The improvement achieved by DualAug is reasonable, compared to the previous method.
>
> | Method | Accuracy(%) | $\Delta$ |
> | :---: | :---: | :---: |
> | AutoAugment (2018) | 82.90 | - |
> | RandAugment (2019) | 83.30 | 0.40 |
> | TrivialAugment (2021) | 83.54 | 0.24 |
> | DeepAA (2022) | 84.02 | 0.48 |
> | DeepAA+DualAug (2023) | **84.39** | 0.37 |
>
> Moreover, as observed by the reviewer, DualAug's **"stable on different datasets"** is quite noticeable. In addition to different datasets, it is also stable across various baseline data augmentations (AA, RA, and DeepAA) and tasks (classification, semi-supervised learning, and self-supervised learning). Our extra experiment on more dataset in Response to Reviewer c5kH (1/7) also proves that DualAug is competitive.

---

> > ### Public Comment · ~Anqi_Xiao1 · 2023-11-23
> > **Questions about DualAug with TrivialAugment**
> >
> > Dear authors,
> >
> > You've proposed a really interesting work that focuses on the OOD, which is essential for data augmentation. I'm interested in your work. However, I think the results of TrivialAugment (TA) using RandAugment space listed in the table above is not quite fair, since one of the important designs of TA is the over-range (the Wide space) for transformations. The performance of TA (Wide) is 84.33%.
> >
> > A worthwhile experiment is the combination of DualAug with TA(Wide). Since TA benefit significantly from the expanded search space, I'm wondering whether DualAug can further filter the outliers of TA if using DualAug. This is important because the OOD capability of TA is lacked while it suffers from the problem of over-transformation.
> >
> > Looking forward to your discussions.

---

> ### Author Response · Authors · 2023-11-22
> **Response to Reviewer WegG (5/5)**
>
> **Question 3: How often are mean and std of the basic distribution estimated?**
>
> The mean and std of the basic distribution are estimated in every iteration. In Appendix A, Algorithm 1 details the process for estimating the mean and std of the basic distribution.
>
> We would like to express our gratitude once again for taking the time to review our manuscript. We hope that my response has addressed your concerns and that you will reconsider the value of my manuscript.

---

> ### Author Response · Authors · 2023-11-23
> **Response to Anqi Xiao**
>
> Dear Anqi Xiao,
>
> &nbsp;
>
> Thanks for your interest of our work and the suggestion of testing with TA (Wide).
>
> TA (Wide) uses the same set of transformations as in TA (RA) but with a wider range of strength. Indeed, it is possible that using such a larger augmentation space could produce more OOD data. Following your suggestion, we will report the combination results where TA (Wide) is adopted as the heavy augmentation branch and TA (RA) is as the basic augmentation branch, though probably after rebuttal (as the experiment is still ongoing).
>
> Moreover, we would like to note that, in our DualAug, we build the heavy augmentation branch by adopting additional transformation operations on the basis of the basic augmentation (e.g., AA, RA, and DeepAA), aiming to exploit the diversity of augmentations possible. Thus, though slightly different from what was suggested, an experiment on TA (Wide) in this setting has been done recently, in which we compared TA (Wide) and TA (Wide) + DualAug on CIFAR-10 using WRN-28-10. The results are shown in the table below, and the statistic is calculated from three independent runs. It can be seen that our DualAug still leads to performance gains.
>
> | Method | Accuracy(%) |
> | --- | --- |
> | TA(Wide) | 84.16±.11 |
> | TA(Wide)+DualAug | **84.48±.13** |
>
> &nbsp;
>
> Best regards,
> Authors.

---

> > ### Public Comment · ~Anqi_Xiao1 · 2023-11-24
> >
> > Thanks for your response. I understand the effectiveness of DualAug, which also provides me with some inspirations. It indeed improves the basic performance of mainstream data augmentation methods, though the efficieny may be further considered for wide applications, since only one branch is chosen to apply while another is wasted.
> >
> > Hope to see your code for further details.

---

### Author Response · Authors · 2023-11-22
**General Response**

We thank all the reviewers for their constructive feedback, valuable suggestions, and meaningful questions, which helped us improve our manuscript. We apologize for the delayed response, as we have been working diligently to address the concerns of the reviewers as comprehensively as possible.
In response to the reviewers' recommendations, we have incorporated the following additional content into our revision:
- DualAug's fairness comparison with TeachAugment and KDforAA
- A thorough examination of the computational cost of DualAug and other similar methods
- DualAug's results on the fine-grained visual classification dataset
- DualAug's results on the object detection task
- An ablation study of DualAug's OOD Score
- DualAug's results based on mixup
- The reference to CUDA and DADA

We sincerely hope that our response has addressed your concerns and that you will reconsider the value of our manuscript. Should you have any additional worries, we are open to further discussion with you.

---

### Meta-Review · Area_Chair_gSax · 2023-12-11

**Metareview:**

In this work, the authors present a new method for data augmentation to improve predictive performance in image classification tasks. In particular, the authors expand on previous augmentation methods (e.g. AutoAugment) in order to provide two different kind of transformations to augment an image. The second augmentation is a heavier form of augmentation combined with an OOD detector to generate augmented samples that may improve the predictive performance of a trained model. The authors provide ablation experiments to demonstrate the necessity of these components in the overall performance of the method. The authors demonstrate the efficacy of their method on image classification with CIFAR-10, CIFAR-100, SVHN, ImageNet and robustness versions of ImageNet. The reviewers commented positively on the simplicity of the approach, the selection of datasets to test their methodology, and the ablations. The reviewers also commented negatively on the added complexity of additional hyperparameters, a lack of comparisons around computation cost and the marginal improvements in performance.

The authors provided some responses but no reviewer increased their score after the rebuttals. More importantly, no reviewers championed the paper for acceptance. Given the lack of champions, I took the opportunity to see if I could provide that endorsement. Upon my read of the paper, I concurred with the reviewers’ concerns. I think that the added complexity of hyperparameters and the heavy handed approach of running a separate inference with another model is problematic and may easily lead to suboptimal behavior. Furthermore, given that the gains are marginal and have only been applied to image classification, I am unclear how much this method would be adopted by the broader community. For these reasons, I can not offer that endorsement as a champion of this paper, and given these concerns the paper will not be accepted in this conference.

**Justification For Why Not Higher Score:**

Marginal gains. Complex method.

**Justification For Why Not Lower Score:**

N/A

---

### Decision · Program_Chairs · 2024-01-16

Reject